# Urinary 5-Hydroxyindolacetic Acid Measurements in Patients with Neuroendocrine Tumor-Related Carcinoid Syndrome: State of the Art

**DOI:** 10.3390/cancers15164065

**Published:** 2023-08-11

**Authors:** Roberta Elisa Rossi, Elisabetta Lavezzi, Simona Jaafar, Giacomo Cristofolini, Alice Laffi, Gennaro Nappo, Silvia Carrara, Alexia Francesca Bertuzzi, Silvia Uccella, Alessandro Repici, Alessandro Zerbi, Andrea Gerardo Antonio Lania

**Affiliations:** 1Gastroenterology and Endoscopy Unit, IRCCS Humanitas Research Hospital, Via Manzoni 56, Rozzano, 20089 Milan, Italy; roberta.rossi@humanitas.it (R.E.R.); silvia.carrara@humanitas.it (S.C.); alessandro.repici@hunimed.eu (A.R.); 2Endocrinology and Diabetology Unit, IRCCS Humanitas Research Hospital, Via Manzoni 56, Rozzano, 20089 Milan, Italy; elisabetta.lavezzi@humanitas.it (E.L.); simona.jaafar@humanitas.it (S.J.); giacomo.cristofolini@humanitas.it (G.C.); 3Hematology and Oncology, IRCCS Humanitas Research Hospital, Via Manzoni 56, Rozzano, 20089 Milan, Italy; alice.laffi@humanitas.it (A.L.); alexia.bertuzzi@humanitas.it (A.F.B.); 4Pancreatic Surgery Unit, IRCCS Humanitas Research Hospital, Via Manzoni 56, Rozzano, 20089 Milan, Italy; gennaro.nappo@hunimed.eu (G.N.); alessandro.zerbi@hunimed.eu (A.Z.); 5Department of Biomedical Sciences, Humanitas University, Via Rita Levi Montalcini 4, Pieve Emanuele, 20072 Milan, Italy; silvia.uccella@hunimed.eu; 6Pathology Service, IRCCS Humanitas Research Hospital, Via Manzoni 56, Rozzano, 20089 Milan, Italy

**Keywords:** neuroendocrine tumors, small intestinal neuroendocrine tumors, carcinoid syndrome, 5-hydroxyindolacetic acid, biomarkers, diagnosis, follow-up, prognosis

## Abstract

**Simple Summary:**

Neuroendocrine tumor (NET)-related carcinoid syndrome (CS) is characterized by symptoms related to hormonal secretion and long-term complications, including carcinoid heart disease (CHD), which is potentially life-threatening. The availability of reliable biomarkers is crucial for the prompt diagnosis and management of these patients. Herein, we summarized available evidence on the role of 24 h urinary 5-hydroxyindolacetic acid (24u5HIAA) as a biomarker for the diagnosis/follow-up of NET-related CS, with particular focus on its potential prognostic role. The 24u5HIAA appears a valuable biomarker for both the diagnosis and management of CS, providing satisfactory sensitivity and specificity. High 24u5HIAA levels correlate with a dismal prognosis, namely an increased likelihood of CHD development and disease progression/mortality. The downside of 24u5-HIAA is represented by the need for a 24 h urine collection, which is unpopular among patients. However, although other interesting biomarkers have been proposed, 24u5HIAA remains the most accurate CS biomarker.

**Abstract:**

Carcinoid syndrome (CS), mostly associated with small intestinal neuroendocrine tumors (SI-NETs) or lung-related NETs, is characterized by symptoms related to hormonal secretion and long-term complications, including carcinoid heart disease (CHD), which is potentially life-threatening. In the early stages of the disease, symptoms are non-specific, which leads to delayed diagnoses. The availability of reliable tumor markers is crucial for a prompt diagnosis and proper management. This review summarizes available evidence on the role of 24 h urinary 5-hydroxyindolacetic acid (24u5HIAA), which is the urinary breakdown metabolite of serotonin, in the diagnosis/follow-up of NET-related CS, with a focus on its potential prognostic role, while eventually attempting to suggest a timeline for its measurement during the follow-up of NET patients. The use of 24u5HIAA is an established biomarker for the diagnosis of NETs with CS since it shows a sensibility and specificity of 100% and 85–90%, respectively. The downside of 24u5-HIAA is represented by the need for 24 h urine collection and the risk of confounding factors (foods and medication), which might lead to false positive/negative results. Moreover, 24u5HIAA is useful in the follow-up of NETs with CS since a shorter double time correlates to a higher risk of disease progression/disease-specific mortality. Furthermore, an elevation in 24u5-HIAA is correlated with a dismal prognosis because it is associated with an increased likelihood of CHD development and disease progression/mortality. Other potentially interesting biochemical markers have been proposed, including plasmatic 5HIAA, although further standardization and prospective studies are required to define their role in the management of NETs. Meanwhile, 24u5HIAA remains the most accurate CS biomarker.

## 1. Introduction

Neuroendocrine neoplasms (NENs) encompass a heterogeneous spectrum of diseases, sharing morphologic and immunophenotypic features of neuroendocrine cells, and include two main categories with distinct clinical and biological features: well-differentiated neuroendocrine tumors (NETs) and poorly differentiated neuroendocrine carcinomas (NECs) [1]. NETs originating from the distal small intestine (SI) (jejunum and ileum) may result in functional syndrome due to their ability to synthesize and secrete bioactive molecules, including serotonin (5-hydroxitriptamine, 5HT), as well as histamine, kallikrein, prostaglandins, and tachykinins. NETs at other sites, such as pulmonary, appendiceal, colonic, and, rarely, pancreatic NETs may also be composed of 5HT-producing cells. Here, 5HT is a tryptophan-derived amine that is physiologically synthesized and stored in normal enterochromaffin-like cells (ECL) in the gastrointestinal (GI) tract, which represents the main source of this molecule in the body (80% of total body content). The remaining amount is produced by neurons in the central nervous system and by platelets. The increased secretion of 5HT by neoplastic cells, mainly the SI-NETs, is the main driver of carcinoid syndrome (CS) and is defined by several symptoms, including secretory diarrhea, flushing, bronchospasm, and long-term fibrotic changes in the mesentery and in the cardiac valves, thereby potentially leading to carcinoid heart disease (CHD), which is potentially life-threatening [2]. The 5HT is synthesized by the uptake of the essential amino acid tryptophan (TRP) or is ingested from the diet and metabolized by monoamine oxidases (MAO) in the liver, lungs, and brain to 5-hydroxyindolacetic acid (5HIAA), which is excreted by the kidney (Figure 1). In more detail, the initial step involves the conversion of TRP to the short-lived 5-hydroxytryptophan (5HTP) by the enzyme L-tryptophan-5-hydroxylase (TPH). The subsequent metabolic step involves the conversion of 5HTP to 5HT by the cytosolic enzyme aromatic acid decarboxylase. A further step involves the MAO, which converts 5HT to 5-hydroxyindole acetaldehyde (5HIA), which is finally metabolized to 5HIAA—the major excreted metabolite of 5HT (Figure 2). In healthy subjects, approximately 1% of the dietary TRP is converted to 5HT. This value increases to 70% in patients with CS, due to inappropriately high levels of the biologically active 5HT metabolite that reaches the peripheral circulation by bypassing the first metabolic step in the liver, which normally inactivates these products [3,4]. This usually happens in patients with hepatic metastatic disease, or when the portal vein is bypassed (e.g., ovarian or bronchial primaries) [5].

In the early stage of CS, symptoms are non-specific, which leads to delayed diagnoses. Therefore, the availability of reliable tumor markers is crucial for a prompt diagnosis of NET patients. In this setting, the assessment of 24 h urinary 5HIAA (24u5HIAA) excretion has been shown to be a valuable tool for both the diagnosis and management of patients with CS; of note, it correlates with the severity of CHD. Other potentially interesting biochemical markers have been proposed, including plasmatic 5HIAA, although further standardization is required to clearly define their role in the management of NET patients. On the other hand, the utility of 24u5HIAA (and maybe plasma 5HIAA levels) as a prognostic factor for survival in SI-NETs is still a matter of debate.

Based on the above observations, the current review summarizes the available evidence of the role of 24u5HIAA in the diagnosis and follow-up of NET-related CS, with a specific focus on its potential prognostic role, and eventually attempting to suggest a timeline for its measurement during the follow-up of NET patients, according to data from the literature and our own experiences.

## 2. Materials and Methods

A bibliographical search was performed in PubMed to identify guidelines and primary literature (retrospective and prospective studies, systematic reviews, and case series) published in the last 15 years, using both medical subject heading (MeSH) terms and free-language keywords: neuroendocrine tumors, small intestinal neuroendocrine tumors, carcinoid syndrome, 5-hydroxyindolacetic acid, biomarkers, diagnosis, follow-up, and prognosis. The reference lists from the studies returned following the electronic search were manually examined to identify further relevant reports. The reference lists from all available review articles, primary studies, and proceedings of major meetings were also considered. Articles published as abstracts were included, whereas non-English language papers were excluded.

## 3. Results

A total of 133 records were reviewed and 50 were considered pertinent for the purpose of the present review.

### 3.1. Diagnostic Accuracy of 24u5HIAA for Carcinoid Syndrome

According to the latest European Neuroendocrine Tumor Society (ENETS) guidelines, 24u5HIAA should be measured at presentation in all patients with advanced SI-NETs, in lung/ovary NETs of any stage, in patients with unknown primary origin (UPO) NETs with liver metastases, and in every NET patient with suspected CS [6].

Other well-known general biomarkers for NETs include Chromogranin A (CgA) and neuron-specific enolase (NSE). However, these are characterized by poor diagnostic accuracy and a high percentage of false positive and/or negative results [7]. Indeed, it is important to highlight that CgA is not very useful in a diagnostic setting, whereas it might provide useful data in the follow-up of NET patients [8]. Furthermore, CgA seems to have a prognostic role since it has been reported to be closely related to the degree of tumor differentiation and tumor burden [9,10]. However, these are considered to be general biomarkers, and in the suspicion of CS, 24u5HIAA is still considered the most accurate biomarker. A comparison between CgA and 24u5HIAA is reported in Table 1.

The measurement of 24u5HIAA is useful in the differential diagnosis of patients developing symptoms representative of CS. The diagnosis of CS mostly derives from clinical suspicion. However, other causes of flushing (e.g., menopause-related changes) and diarrhea should be first excluded [14]. Diarrhea is very common in NET patients but it is not always a result of CS as it can result from other more common GI disorders (e.g., inflammatory bowel disease and irritable bowel syndrome), bile malabsorption, frequent surgical sequelae, somatostatin analog and other treatments, exocrine pancreatic insufficiency [15]. In the presence of symptoms suggestive of CS, a 24u5HIAA secretion above 50 μmol is considered compatible with a diagnosis of CS [6]. However, in the neuroendocrine setting even more than in other contexts, radiological tools, including computed tomography (CT) scan or magnetic resonance imaging (MRI), together with positive somatostatin receptor imaging (SRI), namely Gallium-68 Positron Emission Tomography (PET) scans are necessary to detect primary tumors and associated metastases [6,16]. A recent retrospective study [16], analyzing 22 of 196 patients with carcinoid-like symptoms and no evidence of primary NET based on anatomical imaging and endoscopy, reported that the Gallium-68 PET scan was particularly useful in detecting NETs in symptomatic patients with negative anatomical imaging and can change the treatment in these patients.

#### 3.1.1. Sensitivity and Specificity

The accuracy of 24u5HIAA depends greatly on the clinical context. In the presence of CS, the overall sensitivity and specificity of 24u5HIAA are in the order of 70% and 90%, respectively [17,18]. In functional SI-NETs, discriminating performances can vary depending on whether the cut-offs are high or low. According to Meijer et al., a low 24u5HIAA cut-off value (2.8 mmol/mol creatinine) yielded 68% sensitivity and 89% specificity, whilst a higher cut-off (6.7 mmol/mol creatinine) improved specificity to 98%, although at the expense of lower sensitivity (52%) [17]. Therefore, to confidently exclude a SI-NET, a low-level cut-off value should be preferred; conversely, to confirm the presence of SI-NETs, a high-level cut-off value is considered to be better. Since 24u5HIAA is eliminated in the urine, a loss of renal function could decrease its urinary excretion, leading to an increase in plasma concentrations, which could have an impact on the specificity of both plasma and urinary tests.

#### 3.1.2. False Positive and False Negative Results

Despite having high specificity, false-positive results may be induced by the ingestion of certain drugs and tryptophan/serotonin-rich foods (Table 2). Therefore, patients should abstain from these foods and drugs for 3 days prior to and during the urinary collection [19]. Other false-positive results could be determined by underlying GI disorders (e.g., celiac disease, Whipple disease, and cystic fibrosis), and characterized by changes in 5HT degradation or metabolite excretion [17].

On the other hand, false negative results are also possible in patients with CS, particularly in the absence of diarrhea [20]. Additionally, the possibility of CS being associated with normal 5HIAA levels could be explained by the presence of other circulating biologically active molecules, which may be often secreted or co-secreted [5]. False-negative results are also associated with certain drugs (Table 3). As previously stated, 24u5HIAA levels could be falsely low in patients with chronic kidney disease or in those on hemodialysis [5].

#### 3.1.3. Collecting Procedure

In order to obtain an exact 24 h collection, collecting should start at a defined time point following urination, and after that urine should be collected until the same time point the following day. Urine should be collected and measured in plastic containers and acid should be added to ensure sterility and stability. The addition of a weak acid, such as acetic acid, was recommended but it was found to interfere with the colorimetric method [21]. Commonly, hydrochloric acid is used to lower the pH to 3. The sample should be stored in a refrigerator until analysis. All the urine passed over 24 h should be collected into the container, preferably by using a measuring jug. In order to avoid sampling errors, written instructions should be handed out, including for food and medication precautions [22]. Significant intra-individual variations in 24u5HIAA should be considered, thus, two consecutive 24 h collections should be performed and the mean value of these two can be taken [5].

It is a common experience that the collection of 24u5HIAA is quite complicated and inconvenient, with several factors potentially affecting the accuracy of this procedure. These include forgetting to collect some of the urine, over exceeding the 24 h collection period and collecting more urine than requested, losing urine from the specimen container through spillage, and not keeping urine cold during collection. Although in the literature there is a paucity of studies addressing the difficulty of collecting 24 h urine samples in the outpatient setting, this test is known to be unpopular among patients [5]. Therefore, there is a need for different, possibly more pragmatic, biochemical markers; however, none have yet been standardized in clinical practice for the diagnosis and management of NET-related CS.

#### 3.1.4. Plasmatic 5HIAA and Overnight Urinary 5HIAA

Adaway et al. compared 5HIAA concentrations in paired serum and plasma samples, then analyzed paired urine and serum samples in 134 subjects (108 patients with known NETs and 26 healthy volunteers) [23]. The authors demonstrated that plasma and 24u5HIAA measurements had similar diagnostic accuracies. The use of plasmatic 5HIAA (p5HIAA) would presumably be preferred by the patient since collecting all the urine passed over 24 h can be challenging and time-consuming, with many specimens either being over- or under-collected, even when patients are correctly educated. Additionally, plasma measurements are less severely affected by serotonin-containing foods. A 24 h dietary restriction is sufficient to avoid the risk of interference, rather than the 72 h restriction, which is currently recommended for the 24 h urine collections. However, the plasma measurement should be restricted in patients with normal renal function (<60 mL/min/1.73 m^2^) as a loss in renal function could lead to falsely increased plasma concentrations of 5HIAA. The strong linear correlation between the measurements of urinary and serum 5HIAA was confirmed in a recent study [24], where a concordance of 89% was found in the presence of a normal renal function [24]. A prospective multicenter study [25] compared the diagnostic performances of 24u5HIAA, overnight urinary 5HIAA (Ou5HIAA), and p5HIAA in midgut NETs. According to the results of this study, the reproducibility of 24u5HIAA, Ou5HIAA, and p5HIAA was reported to be excellent, with significant discrimination between patients and controls with irritable bowel syndrome; furthermore, these biomarkers were associated with CS, CHD, and tumor burden independently of other prognostic factors. Moreover, the levels were similar even in the presence of suboptimal diet observance. Both Ou5HIAA and p5HIAA could be considered convenient alternatives to 24u5HIAA for patients with midgut NETs. However, published experience is confined to a small number of reports and further studies are needed to draw more solid conclusions. As reported by current guidelines [6], p5HIAA is an alternative method that can effectively differentiate CS patients from controls and correlates with the presence of CS and CHD; however, its availability is limited and its use is not routinely recommended.

### 3.2. Performance of 24u5HIAA in Follow-Ups

#### 3.2.1. 24u5HIAA and CHD

About 50% of the patients with CS will develop CHD, which can be life-threatening [6,26,27]. The secretion of vasoactive peptide (in particular 5HT) is thought to cause the deposition of cardiac plaques, which occur most commonly in the valves and endocardium on the right side of the heart (~90% of the cases). Lung metabolism prevents the involvement of the left heart by inactivating serotonin and other tumor products. However, the pulmonary degradative capacity might be overwhelmed in the presence of very high levels of serotonin, potentially leading to left-sided valve involvement, which can also occur when vasoactive peptides skip lung metabolism, as in the presence of a congenital heart defect known as patent foramen ovale or in a functioning lung NET. A recent meta-analysis [27] showed that elevated levels of 24u5HIAA are predictive of an increased risk of CHD, although there is no absolute cut-off value to predict the development of CHD. It is for this reason that, even in the absence of symptoms suggestive of CS, it is suggested to periodically measure 24u5HIAA in most patients with metastatic midgut NETs. However, according to ENETS guidelines, 24u5HIAA > 300 μmol confers a 2- to 3-fold increase in the risk of CHD development/progression [6]. Patients identified as at high risk of CHD should be promptly referred to cardiologists and undergo echocardiogram screening. Furthermore, as suggested by the latest ENETS guidelines, the evaluation of the amino-terminal pro-brain natriuretic peptide should be added to all patients with elevated 24u5HIAA in order to increase the accuracy of the risk prediction [6].

#### 3.2.2. Effect of Treatments on 24u5HIAA Levels

All patients with CS should be treated as soon as the diagnosis has been made. The control of hormone hypersecretion is evidenced by reduced or normalized levels of 24u5HIAA, although data on 24u5HIAA changes associated with specific treatments are limited.

The first-line therapy for CS is represented by somatostatin analogs (SSAs) [28,29,30,31], which bind to somatostatin receptors (SSTRs) and inhibit tumor secretion, ultimately, improving symptoms in up to 70% of CS patients and decreasing 24u5HIAA levels in approximately 46%.

In patients with recurring or persisting CS symptoms and increasing or persistently high 24u5HIAA levels, despite the use of maximum-label doses of SSAs, a careful differential diagnosis should be taken into account. First, it is important to exclude other causes of diarrhea [15] and 24u5HIAA false positive results. Furthermore, issues with SSA administration should be assessed, as decreased SSA absorption relating to fibrosis at the injection site may develop in patients receiving long-term SSA. If CS is controlled only immediately after the SSA injection, dose escalation can also be considered. Overall, SSA dose escalation was demonstrated to improve symptoms in nearly 80% of the cases, while only 29% of patients showed further reduction in 24u5HIAA levels [32].

Minimal evidence is available on what is the best therapeutic sequence for refractory CS [6].

Among the possible therapeutic choices in patients with persistent symptoms, despite optimal treatment with SSA, peptide radionuclide receptor therapy (PRRT) with 177Lu-DOTA0-Tyr3-Octreotate (177Lu-DOTATATE) is the most efficient treatment in the presence of positive SRI [33]. PRRT has been proven to provide a clinically robust quality-of-life benefit and to significantly decrease 24u5HIAA [34]. Zandee et al. described the effects of treatment with 177Lu-DOTATATE in 22 patients with non-progressive midgut NETs and refractory CS and showed that PRRT was a viable, safe, and effective option, which was able to reduce 24u5HIAA excretion by more than 30% in 56% of the patients [35].

The last drug approved by the Food and Drug Administration and European Medicines Agency for the treatment of refractory CS was telotristat ethyl, an oral inhibitor of TPH, which represents the rate-limiting step in serotonin biosynthesis [36]. It has long been known that the inhibition of TPH could alleviate CS-related symptoms and decrease urinary 5HIAA levels [37] in the CS refractory to SSAs [36,38,39,40]. In two phase III clinical trials, TELESTAR and TELECAST [39,40], a significant favorable effect was reported for telotristat ethyl use on diarrhea. Of note, in these patients at week 12, the mean 24u5HIAA levels decreased by 40 mg and 57.7 mg per 24 h with telotristat at doses of 250 mg and 500 mg, respectively, while levels increased in the placebo group by 11.5 mg per 24 h in the same time period [39]. Telotristat ethyl might also have antiproliferative effects but no clear evidence of such is available [41]. According to the latest ENETS guidelines, telotristat ethyl has a major role in the management of refractory diarrhea predominant-CS [6].

Everolimus, an oral inhibitor of mTOR, is registered as an antiproliferative treatment option for progressive NETs but has been shown to also have antisecretory properties in patients with CS, as demonstrated in the RADIANT 2 phase III trial [42]. In particular, the combination of everolimus and octreotide reduced 24u5HIAA (defined as normalization, or a 50% or greater reduction from the baseline values) more than octreotide alone (85 out of 140 [61%] vs. 76 out of 141 [54%]; *p* < 0.0001). Limited data exists on the effects of everolimus on CS-related symptoms. In a small retrospective study, approximately 70% of patients with CS had an improvement in some or all symptoms after the addition of everolimus to octreotide [43]. The addition of everolimus to SSA therapy has a limited role but it can be considered in the treatment of refractory CS [6,28].

Chemotherapy has no clear indication in the treatment of CS [6]. The majority of data on chemotherapy are old and inconsistently report CS-specific outcomes, restricted only to a 5HIAA response. In particular, a 24u5HIAA response rate of 31% has been described in 111 patients following different chemotherapeutical regimens [28].

Table 4 summarizes the effects of available treatment options for CS on 24u5HIAA levels.

### 3.3. Performance of 24u5HIAA as a Prognostic Marker

Patients with SI-NETs have heterogeneous survival. Several studies have investigated prognostic factors, identifying some unfavorable markers for survival, such as high plasma CgA levels, the presence of liver or lymph node metastases, CHD, tumor size, histological grade of differentiation, and elderly age. Currently, the usefulness of 24u5HIAA as a prognostic marker in patients with SI-NETs is not clear [27,44].

Laskaratos et al. evaluated prognostic factors for overall survival (OS) in a large cohort of patients with SI-NETs and mesenteric desmoplasia [12]. The authors found that in patients over 65, a baseline 24u5HIAA more than 10 times the upper limit of normal was predictor of poor OS by both univariate analysis and multivariate analyses, while persistently low 24u5HIAA was associated with increased OS. Similarly, Van Der Horst-Schrivers et al. demonstrated that the 24u5HIAA level is an independent prognostic factor at both the referral and every moment during the follow-up for patients with a disseminated midgut carcinoid tumor [13]. Individuals with a significantly increased 24u5HIAA level at referral (>20 mmol/mol creatinine) had a median survival of 33 months compared to 90 months for patients with a moderately increased level (≤20 mmol/mol creatinine). On the other hand, in a study by Zandee et al., 24u5HIAA levels greater than 10 times the upper limit of normal levels were associated with poor OS in a univariate model for patients with midgut NETs; however, other tumor markers, including CgA and NSE, and tumor grade were far more powerful predictors and 24u5HIAA lost its prognostic significance in the multivariate analysis [44].

In a recent retrospective cohort study of 184 patients with G2 SI-NETs treated with SSAs, interferon-alpha (IFN), or PRRT [45], the authors reported that baseline 24u5HIAA was associated with cancer-specific survival (CSS) following treatment with IFN and PRRT, yet not for SSA treatment alone and that it was a statistically significant prognostic marker of progression free-survival (PFS) for patients treated with PRRT. Suggested cut-offs were 6 × the upper limit of normal levels for both CgA and 5HIAA in CSS. Moreover, early reductions (i.e., after 6 months of treatment) in CgA and 5HIAA were reported to be prognostic for treatment with SSA but not PRRT.

The 24u5HIAA value is of particular relevance in the context of CHD. According to a recent systematic review [27], high levels of 24u5HIAA have been shown to be positively correlated with an increased probability of CHD diagnosis, disease progression, and mortality. In detail, 24u5HIAA > 300 μmol confers a 2- to 3-fold increase in the risk of CHD development/progression [6]. On the other hand, it is currently unclear if one urinary assessment with non-elevated 5HIAA excretion is sufficient to rule out the development of CHD.

The use of 24u5HIAA as a marker of disease progression has also been proposed. A single-center study evaluated 48 patients with NETs and concurrent serum and urinary 5HIAA testing, as well as CT/MRI [24]. However, no correlation was evident between RECIST 1.1 responses [46] and changes in serum 5HIAA levels, suggesting that its role as a marker of disease progression is still limited [24].

## 4. Discussion

CS is the most common functional syndrome associated with well-differentiated NETs, which affects the patient’s prognosis and survival, also due to the possible occurrence of CHD [6]. Since symptoms are non-specific, the availability of reliable tumor markers is crucial for both a prompt diagnosis and subsequent management of patients.

The urinary breakdown metabolite of 5HT is urinary 5HIAA as its levels serve as an indicator of 5HT production. As indicated by the latest ENETS guidelines, clinicians should measure 24u5HIAA in advanced SI-NETs, in lung/ovary NETs of any stage, in UPO-NETs with liver metastases, and in every NET patient, whenever there is the suspicion of CS [6]. In the setting of SI-NETs associated with CS, 24u5HIAA has very good diagnostic accuracy when confounding factors are avoided (see Table 2 and Table 3) [5] and 24u5HIAA secretion above 50 μmol is generally considered diagnostic for CS [6]. Although the collection of 24u5HIAA is quite complicated, making this test unpopular among patients [5], no other biomarkers are currently available in real-life clinical practice for the diagnosis and follow-up of NET-associated CS.

The implementation of 24uHIAA may also represent a useful tool for the follow-up of NETs with CS. The ENETS guidelines recommend its measurement for follow-up in patients with lung, colon, and appendiceal NETs if elevated at diagnosis [47]. Tirosh et al. found that patients with a shorter 24u5HIAA doubling-time (DT) had a higher risk of disease-specific mortality, and mainly accounted for the subgroups of patients with SI-NETs or UPO-NETs. In addition, a short 24u5HIAA DT in patients with SI-NETs or UPO-NETs was associated with a higher risk of disease progression, in both univariate and multivariable analyses [11]. However, further validation in an independent cohort would be necessary.

In patients with CS, which was complicated by CHD, a recent meta-analysis [27] showed that increased levels of 24uHIAA were associated with CHD, disease progression, and mortality [6], whereby levels > 300 μmol were associated with a 2- to 3-fold increase in the risk of CHD development/progression [6].

When CS is diagnosed, prompt medical treatment should be started [6]. Available treatment options are aimed at controlling hormone hypersecretion, reducing the tumor burden, and alleviating symptoms. As a matter of fact, all treatments, particularly if effective, reduce 24u5HIAA levels; on the other hand, it is not clear whether a significant increase in 24u5HIAA levels during the follow-up, without a corresponding radiological tumor progression, might be necessary for a treatment escalation. However, this should be discussed case by case by a multidisciplinary team, taking into account the disease severity, the presence of CHD, tumor status, and individual factors [47].

Even if the role of 24u5HIAA as a prognostic marker in SI-NETs is not yet fully clarified, its use might have some potential implications in the management of patients with CS. Previous studies have correlated 24u5HIAA levels at presentation with outcome in the univariate analysis, yet after correction for other factors such as age, it did not stand out [48,49,50]. Recent trials, instead, have demonstrated that the 24u5HIAA level is an independent predictor of poor OS in both univariate and multivariate analyses [13,24]. In particular, Van Der Horst-Schrivers et al. demonstrated for the first time that the 24u5HIAA level is an independent prognostic factor both at referral and at every moment during follow-up for patients with a disseminated midgut carcinoid tumor, using a cut-off of >20 mmol/mol creatinine and associated with a dismal prognosis [13].

These findings might have potential clinical implications. Since 24u5HIAA levels are prognostic at every moment during the follow-up, it is important to measure 24u5HIAA at every following check-up together with radiological examinations. In the case of increased levels without a corresponding radiological tumor progression, the case should be discussed by a multidisciplinary team to decide whether the increase in the biomarker might justify starting (or changing) a treatment. However, as it is common in everyday clinical practice, this disagreement between values of biomarkers values and radiology could determine, instead, a closer follow-up from both a radiological and cardiologic point of view. Based on available data and on our own center experience, we might suggest testing for 24u5HIAA every 6 months in patients under treatment, particularly with SSA, and also to consider that a 6-month reduction in CgA and 24u5HIAA has been reported as prognostic for treatment with SSAs [45]. Furthermore, in those cases where there is a significant increase in the 24u5HIAA levels, although the disease appears to be steady at radiology, a closer follow-up, including a Gallium68 PET scan together with the measurement of 24u5HIAA, every 3 to 6 months and a complete cardiologic examination, including an echocardiogram, is suggested. However, it is important to highlight that, independently of the 24uHIAA levels, once a diagnosis of CS has been established, the referral of the patient to a cardiologist with specific expertise in this field is mandatory.

## 5. Conclusions

In summary, 24u5HIAA is a reliable marker, which is complementary to radiology and nuclear medicine tools, for the diagnosis of CS and potentially provides relevant prognostic data, particularly in correlation with CHD. Therefore, it is important to measure 24u5HIAA levels if there is a suspicion of NET-related CS, to confirm the diagnosis, although its actual role as a prognostic marker is not fully clarified. Thus, according to available data, we suggest that it should be repeatedly measured during the follow-up in order to modify the timeline of the follow-up itself and/or the treatment strategy, accordingly, on a case-by-case basis and always in a multidisciplinary setting.

## Figures and Tables

**Figure 1 cancers-15-04065-f001:**
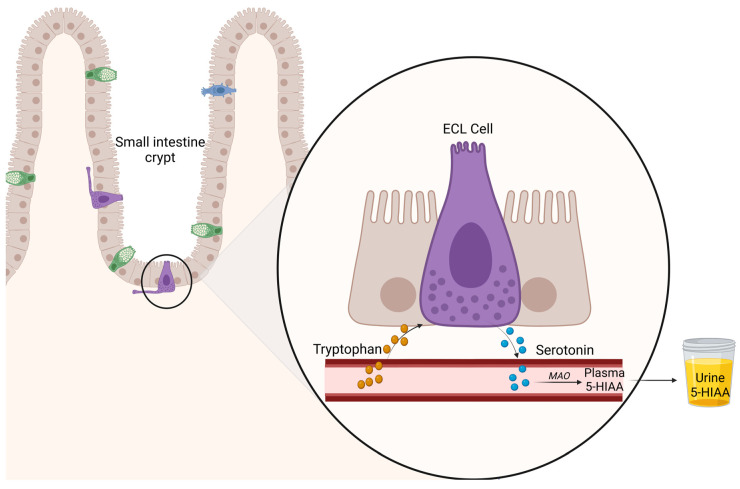
Serotonin pathway in small intestinal enterochromaffin-like cells. ECL cell: enterochromaffin-like cell; MAO: monoamine oxidase; 5HIAA: 5-hydroxyindolacetic acid. Created using BioRender.com.

**Figure 2 cancers-15-04065-f002:**
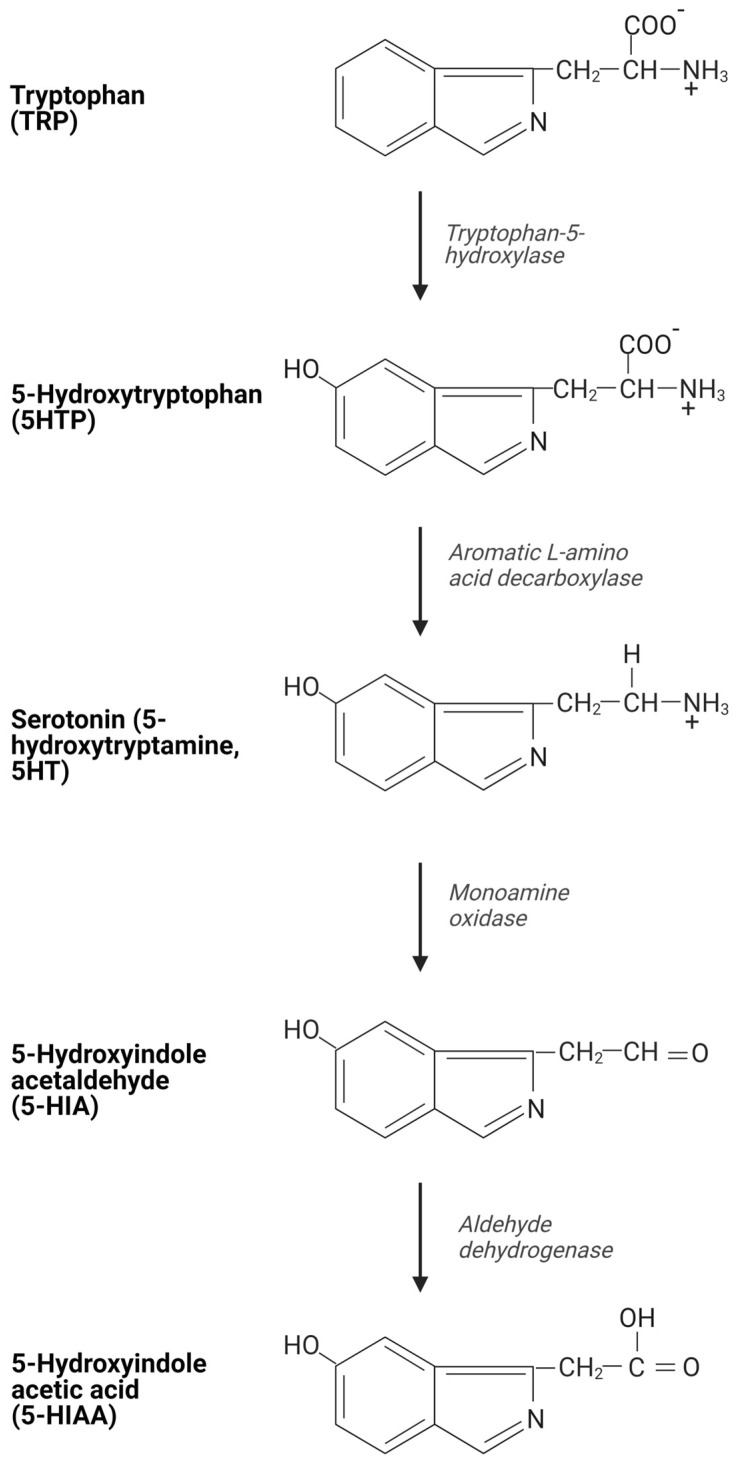
The biochemical pathway for serotonin synthesis and metabolism. Created using BioRender.com.

**Table 1 cancers-15-04065-t001:** Comparison between Chromogranin A and 24 h urinary 5-hydroxyindolacetic acid.

Biomarker	Type	Diagnosis	Follow-Up	Prognosis
CgA	Monoanalyte	Sensitivity 67–93%, specificity > 85% for all NETs [5]	Reliable marker to monitor disease progression, response to treatment, and the early detection of recurrence after treatment [8,10]	Elevated CgA (>6 × upper normal limit) has been associated with worse outcomes and CgA correlates with tumor differentiation and burden [10]
24u5HIAA	Monoanalyte	Sensitivity of 70% and specificity of 90% for NET-related CS [5]	If elevated at diagnosis, 24u5HIAA should be measured in the follow-up: patients with shorter DT have a higher risk of disease progression and disease-specific mortality [6,11]	High 24u5HIAA values correlate with poor OS [12,13]. Values of 24u5HIAA > 300 μmol confer a 2- to 3-fold increase in risk of CHD development/progression [6].

CgA: Chromogranin A; CHD: carcinoid heart disease; CS: carcinoid syndrome; DT: doubling time; OS: overall survival; NETs: neuroendocrine tumors; 24u5HIAA: 24 h urinary 5-hydroxyindolacetic acid.

**Table 2 cancers-15-04065-t002:** Factors interfering with measurements of urinary 5-hydroxyindole acetic acid: factors producing false-positive results. Adapted from ENETS Consensus Guidelines for the standard of care in Neuroendocrine Tumours: Biochemical Markers. 2017.

Factors Producing False-Positive Results	
Foods	Drugs
Avocado	Acetominophen
Banana	Acetanilid
Chocolate	Caffeine
Coffee	Fluorouracil
Eggplant (aubergine)	Guaifenesin
Pecan	L-DOPA
Pineapple	Melphalan
Plum	Mephenesin
Tea	Methamphetamine
Walnuts	Methocarbamol
	Methysergide maleate
	Phenmetrazide
	Reserpine
	Salicylates

**Table 3 cancers-15-04065-t003:** Factors interfering with measurements of urinary 5-hydroxyindole acetic acid. Factors producing false-negative. Adapted from ENETS Consensus Guidelines for the standard of care in Neuroendocrine Tumours: Biochemical Markers. 2017.

Factors Producing False-Negative Results	
Foods	Drugs
None	Corticotrophin
	p-Chlorophenyl alanine
	Chlorpromazine
	Heparin
	Imipramine
	Isoniazid
	Methenamine maleate
	Methyldopa

**Table 4 cancers-15-04065-t004:** Effects of available treatment options for carcinoid syndrome on the levels of the 24 h urinary 5-hydroxyindolacetic acid.

	Study	Reference	No of Patients	Treatment	Symptom Control (% of Patients)	Reduction of 24u5HIAA
SSAs	Phase III (control arm) RADIANT-2	(Pavel et al., 2011) [42]	211	Octreotide LAR 30 mg q. 28 days	27	Normalization or a ≥ 50% reduction from baseline values in 54% of patients
PRRT	Retrospective cohort study (177Lu-Dotatate)	(Zandee et al., 2021) [35]	22	177Lu-DOTATATE 4 cycles up to a cumulative intended dose of 27.8 to 29.6 GBq	67 (flushing) 47 (diarrhea)	Decrease in 24u5HIAA of more than 30% in 56% of patients
TELOTRISTAT	TELESTAR	(Kulke et al., 2017) [39]	135	Placebo tid (45 patients) 250 mg tid (45 patients) 500 mg tid (45 patients)	17 (diarrhea) 29 (diarrhea) 35 (diarrhea)	Mean 24u5HIAA levels decreased by 40 mg and 57.7 mg per 24 h with telotristat 250 mg and 500 mg, respectively
	TELECAST	(Pavel et al., 2018) [40]	76	Placebo tid (26 patients) 250 mg tid (25 patients) 500 mg tid (25 patients)	0 (diarrhea) 40 (diarrhea) 40 (diarrhea)	
EVEROLIMUS	RADIANT-2	(Pavel et al., 2011) [42]	429	Everolimus 10 mg q.d. plus Octreotide LAR 30 mg q. 28 days (216 patients) placebo (213 patients)	NR	Normalization or a ≥ 50% reduction from baseline values: 61% (everolimus plus octreotide) vs. 54% (octreotide alone)
CHEMOTHERAPY	Systematic review and meta-analysis	(Hofland et al., 2019) [28]	-	-	NR	24u5HIAA response rate of 31% has been described in 111 patients following different chemotherapeutical regimens

NR: not reported; PRRT: peptide radionuclide receptor therapy; q.d: once a day; SSAs: somatostatin analogs; tid: three times a day; 24u5HIAA: 24 h urinary 5-hydroxyindolacetic acid.

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
