# Peer review of "Urinary 5-Hydroxyindolacetic Acid Measurements in Patients with Neuroendocrine Tumor-Related Carcinoid Syndrome: State of the Art"

_cancers, 2023, doi:10.3390/cancers15164065_

Round 1
Reviewer 1 Report
This manuscript represents an overview of the use of urine 5-hydrocyindolacetic acid.
- I would like to hear more about other biomarkers used in clinical practice and compare them with urinary 5 hydrocyindolacetic acid. pros/cons, when and why
- how about plasmatic and overnight urinary 5HIAA? INformation given in the article is very limited.
- references: Louis de Mestier and a GTE study (2021)
- I would also appreciate a table to compare 5HIAA with other tools in certain settings (diagnostic,in the follow-up, effects of treatment, etc)
- a state of the art on urinary 5HIAA while omitting other bimarkers or radiographic examinations seems very limited nowadays
Im not a native speaker but minor editing is required
Author Response
Milan, 2023-07-25
Dear Editor,
Thank you for your interest in considering a revised version of our manuscript.
Please find herewith enclosed a copy of our revised manuscript entitled “Urinary 5-Hydroxyindolacetic Acid Measurements in Patients With Neuroendocrine Tumor- related Carcinoid Syndrome: State of Art”
Thank you very much for going into depth with our paper and for your constructive criticism that substantially improved the paper. We believe that all concerns have been successfully rebutted.
All changes have been highlighted in the text, as requested in the instructions for the authors.
We hope the revised version will now be suitable for publication in the Cancers.
Here is our point-by-point response to your comments:
Reviewer #1:
This manuscript represents an overview of the use of urine 5-hydrocyindolacetic acid.
-I would like to hear more about other biomarkers used in clinical practice and compare them with urinary 5 hydrocyindolacetic acid. pros/cons, when and why
Many thanks for this interesting observation. A paragraph re. comparison between 5HIAA and other biomarkers has been added (page 5) together with pertinent references (#7-10).
A referral to CgA and NSE is also reported at page 9: “On the other hand, in a study by Zandee et al., 24u5HIAA greater than 10 times the upper limit of normal was associated with poor OS in a univariate model in patients with midgut NETs [35]; however, other tumor markers, including CgA and NSE, and tumor grade were far more powerful predictors and 24u5HIAA lost its prognostic significance in the multi-variate analysis.”
-How about plasmatic and overnight urinary 5HIAA? INformation given in the article is very limited. - references: Louis de Mestier and a GTE study (2021)
As reasonably suggested, further data has been added (see page 7) together with the pertinent reference (#23)
-I would also appreciate a table to compare 5HIAA with other tools in certain settings (diagnostic,in the follow-up, effects of treatment, etc)
A table reporting a comparison between 24u5HIAA and CgA has been added (Table 1)
-A state of the art on urinary 5HIAA while omitting other bimarkers or radiographic examinations seems very limited nowadays.
We thank the Reviewer for this correct observation and as requested a paragraph re. other biomarkers as well as a referral to radiology have been added (see page 5) with pertinent references (# 7-10 and #14).
Furthermore, a referral to CT and MRI as complementary tools is reported at page 10: “The use of 24u5HIAA as a marker of disease progression has also been proposed. A single center study evaluated forty-eight patients with NETs and concurrent serum and urinary 5HIAA testing, as well as CT/MRI [16]. However no correlation was evident be-tween RECIST 1.1 responses [38] and changes in serum 5HIAA levels, suggesting that its role as a marker of disease progression is still limited [16].
Finally, also in the conclusion, a sentence re. the complementary role of 5HIAA with radiology and nuclear medicine has been added: …”24u5HIAA is a reliable marker, complementary to radiology and nuclear medicine tools”.
-Comments on the Quality of English Language: Im not a native speaker but minor editing is required
As reasonably suggested, English language editing has been performed.
Reviewer 2 Report
This is a comprehensive manuscript addressing 5HIAA in patients with Neuroendocrine Tumors.
Line 29 : "sensibility" is vague. If the authors mean Sensitivity, it may fit better. Line 49 : "synthesize" and...... . Line 53 : tryptophan-derived amine "which is" physiologically ....... . Line 62 : synthesized by "the" essential..... .
Line 65 : "in more detail" rather than in details. Line 70 : by "the" cytosolic.... . Line 70 : "A further step" rather than further step. Line 90 : "The " current ..... . also line 70 summarizes "the " available" ...... .
Figure 1 Good figure but needs more, e.g. the "circle", and the font is small thus difficult to read. Also in Line 94 EC cell, but in Figure 1 ECL cell (please clarify)> Figure 2 is clear to the readers. Line 115 : SI NETs, should be SI-NETS as in the previous pages. This is prevalent in the manuscript, Lines 115, 130, 135, others throughout the manuscript (please check). Line 116 : ; extra space between "patients" and "with". Line 185 should be "increased" rather than increase. Line 196 : "serotonin and the other ...., eliminate "the". Line 199, "patent foramen ovale (a little vague, perhaps can be clarified). Line 200 : in "a" functioning ...., rather than in functioning ....... . Line 208 : add comma after "prediction". Line 223 " comma" after (SSA,). Line224 : "first", rather than firstly. Line 227 :, comma after (assessed,). Line 240: "efficient" rather than efficacious. Line258 : Remove ("a). Line 247 : reported "that" . Line 260 : Comma after daily. Line 264 : " 40mg", and 250mg". Line272 : "also have", rather than have also. Lines and 299: SI-NRTs (please check for others). Line 305 :" Significantly" rather than greatly. Line 319 : CgA and 5HIAA "were" reported ..... . Line 323 : in "detail". Line 321 : reference (did you mean "relevance". Line 334 : did you mean "active" ? Lines 341, 343,351,366, should be "SI-NETs. Line 344 : "factors". Line 375 : 24u5HIAA "is". Line 384 : "center".
I would like to recommend a separate page or two addressing the abbreviations, as there are many of them, and can be confusing.
The English language needs a little more work. Hopefully the changes I suggested will be of help, and will be implemented in the revision of the manuscript.
Author Response
Milan, 2023-07-25
Dear Editor,
Thank you for your interest in considering a revised version of our manuscript.
Please find herewith enclosed a copy of our revised manuscript entitled “Urinary 5-Hydroxyindolacetic Acid Measurements in Patients With Neuroendocrine Tumor- related Carcinoid Syndrome: State of Art”
Thank you very much for going into depth with our paper and for your constructive criticism that substantially improved the paper. We believe that all concerns have been successfully rebutted.
All changes have been highlighted in the text, as requested in the instructions for the authors.
We hope the revised version will now be suitable for publication in the Cancers.
Here is our point-by-point response to your comments:
Reviewer #2:
This is a comprehensive manuscript addressing 5HIAA in patients with Neuroendocrine Tumors.
Many thanks for this appreciation of our paper.
-Line 29 : "sensibility" is vague. If the authors mean Sensitivity, it may fit better.
As reasonably suggested, sensibility has been edited in sensitivity. We do apologize for this typo mistake.
-Line 49 : "synthesize" and.
This has been edited as suggested.
-Line 53 : tryptophan-derived amine "which is" physiologically.
This has been added as properly suggested.
-Line 62 : synthesized by "the" essential.
“The” has been added.
-Line 65 : "in more detail" rather than in details.
This has been edited as suggested.
-Line 70 : by "the" cytosolic.
“The” has been added.
-Line 70 : "A further step" rather than further step.
This has been edited as suggested.
-Line 90 : "The " current.
“The” has been added.
-Also line 90 summarizes "the " available".
“The” has been added as suggested.
-Figure 1 Good figure but needs more, e.g. the "circle", and the font is small thus difficult to read. Also in Line 94 EC cell, but in Figure 1 ECL cell (please clarify)
Figure 1 has been edited according to Reviewer’s suggestion. The abbreviation ECL has been used for both the Figure and the text.
- Figure 2 is clear to the readers.
Many thanks for this kind observation.
-Line 115 : SI NETs, should be SI-NETS as in the previous pages. This is prevalent in the manuscript, Lines 115, 130, 135, others throughout the manuscript (please check).
We thank the Reviewer for this observation and we decided to use SI-NETs throughout the whole manuscript.
-Line 116 : ; extra space between "patients" and "with".
The extra space has been removed.
-Line 185 should be "increased" rather than increase.
This has been edited as reasonably suggested.
-Line 196 : "serotonin and the other ...., eliminate "the".
“The” has been removed.
-Line 199, "patent foramen ovale (a little vague, perhaps can be clarified).
This has been clarified as reported in the text: a congenital heart defect known as patent foramen ovale.
-Line 200 : in "a" functioning ...., rather than in functioning ....
“a” has been added.
-Line 208 : add comma after "prediction".
A comma has been added as suggested.
-Line 223 " comma" after (SSA,).
A comma has been added as suggested.
-Line 224 : "first", rather than firstly.
This has been edited as reasonably suggested.
- Line 227 :, comma after (assessed,).
A comma has been added as suggested.
- Line 240: "efficient" rather than efficacious.
This has been edited as reasonably suggested.
-Line 258 : Remove ("a).
“a” has been removed.
-Line 247 : reported "that" .
As the previous sentence was not fully clear, we edited the verb in described the effects rather than reported the..
-Line 260 : Comma after daily.
A comma has been added as suggested.
-Line 264 : " 40mg", and 250mg".
“mg” has been added as suggested.
-Line 272 : "also have", rather than have also.
This has been edited as reasonably suggested.
-Lines and 299: SI-NRTs (please check for others).
As previously stated, we decided to use SI-NETs throughout the whole manuscript.
-Line 305 :" Significantly" rather than greatly.
This has been edited as reasonably suggested.
-Line 319 : CgA and 5HIAA "were" reported ..... .
This has been edited as reasonably suggested.
- Line 323 : in "detail".
This has been edited as properly suggested.
-Line 321 : reference (did you mean "relevance".
We do apologize for this typo mistake. Reference has been edited in relevance as reasonably suggested.
- Lines 341, 343,351,366, should be "SI-NETs.
As previously stated, we decided to use SI-NETs throughout the whole manuscript.
-Line 344 : "factors".
This has been edited as properly suggested.
-Line 375 : 24u5HIAA "is".
This sentence is “In particular, Schrivers et al. have demonstrated for the first time that 24u5HIAA level is an independent prognostic factor both at referral and at every moment during follow-up for patients with a disseminated midgut carcinoid tumor, being a cut-off of > 20 mmol/mol creatinine associated with a dismal prognosis”
- Line 384 : "center".
This has been edited as properly suggested.
-I would like to recommend a separate page or two addressing the abbreviations, as there are many of them, and can be confusing.
A paragraph including the list of all the abbreviations has been added after the key-words paragraph as suggested (see page 11).
- Comments on the Quality of English Language. The English language needs a little more work. Hopefully the changes I suggested will be of help, and will be implemented in the revision of the manuscript.
Many thanks. The changes suggested were indeed of help in improving the quality of English language.
Reviewer 3 Report
Dear authors for me it is a good manuscript.
We have very few literature of urinary biomarker in cancer yet. So I am happy to read your work. I consider it well detailed in data also in figure than in references.
I suggest only a minor revision because you use two different characters in text editing.
Thank you
Author Response
Dear authors for me it is a good manuscript.
We have very few literature of urinary biomarker in cancer yet. So I am happy to read your work. I consider it well detailed in data also in figure than in references.
I suggest only a minor revision because you use two different characters in text editing.
We do thank the Reviewer for the appreciation of our paper.
The text has been edited as suggested.
Round 2
Reviewer 1 Report
After the changes have been submitted, I have no further recommendations.
I have no further comments.